# Efficient Adaptation of Large Vision-Language Models: Transfer Learning Methods and Applications

## Abstract

Pre-trained large vision-language models (VLMs) have become the dominant choice for handling vision-language tasks, covering from multimodal reasoning to text-image generation. However, these models heavily depend on large-scale training datasets, primarily composed of image-text pairs sourced from web data, which are typically confined to general domains rather than specific downstream tasks. Given the scarcity of data in such specialized domains, transfer learning emerges as a remedy, enabling the adaptation of a model's preexisting knowledge to new tasks with limited data, thereby mitigating the reliance on extensive datasets. Following the current trend of the transfer learning application with vision-language tasks, we provide a systematic study of existing transfer learning techniques adopted for vision-language models, including: (1) a summary of the existing state-of-the-art VLMs, (2) a comprehensive taxonomy of transfer learning approaches for VLMs, (3) the discussion of real-world applications of transfer learning methods for VLMs, (4) a summary of commonly used vision-language dataset and benchmarks in variant vision-language tasks.

## 1 Introduction

Vision-language models (VLMs) present a dynamic frontier in machine learning where the modalities of vision and language intersect to address complex tasks that require the inputs of both modalities. These models process and interpret visual data alongside textual descriptions, enabling machines to understand and generate content that reflects both visual perception and text context. The significance of VLMs lies in their ability to perform a broad spectrum of tasks much better compared to architectures that process modalities independently Shangguan et al. (2025a), such as image captioning (Zhou et al., 2020; Li et al., 2023; Alayrac et al., 2022; Li et al., 2022b; Zhang et al., 2023a), visual question answering (Antol et al., 2015; Kim et al., 2021; Li et al., 2021; Floridi & Chiriatti, 2020), image classification (Radford et al., 2021; Alayrac et al., 2022; Dosovitskiy et al., 2021), etc. These capabilities make VLMs advance numerous practical applications, including robotics system (Chen et al., 2024a; Kuzmenko & Shvai, 2024; Liu et al., 2024b), medical assistance (Chambon et al., 2022; Chen et al., 2022c; Chang et al., 2024), and human sentimental understanding (Wang et al., 2023a; Shi et al., 2024). Despite the potential of VLMs for the broad applications, the development of VLMs that effectively integrate visual and text information to fit in variant domains remains challenging. With the growing trend of pre-training VLMs, model size has grown from 200 million (Li et al., 2021) to 562 billion (Driess et al., 2023) in a few years. At the same time, the size of dataset needed to train the VLMs also has grown from million-scale to billion-scale, making it impractical to develop independent pre-trained VLMs for every sub-domain and every downstream task. Finally, the computational cost to develop a pre-trained VLM has grown to the extent that most of the state-of-the-art VLMs can be only trained by a few large companies.

To address the complexity of the VLM training, transfer learning has emerged as a pivotal strategy in the "Pre-training, Finetuning, Prediction" approach for using VLMs (Xing et al., 2024a). Transfer learning for VLMs involves taking a model pre-trained on one task with massive amount of data, and finetuning it for another specific task using the relatively small amount of data. This approach not only accelerates the training process by utilizing pre-learned features but also enhances the model's ability to generalize from one task to another within or across domains. By transferring knowledge from large, general datasets to

more specialized tasks, VLMs can achieve state-of-the-art performance across a wide range of applications. Currently, Transfer learning and VLMs have been studied with several surveys in each domain. For example, Du et al. 2022 and Zhang et al. Zhang et al. (2024b) survey pretrained VLMs by categorizing them according to the training objective function, architecture, and the used pre-training tasks. Several extensive papers summarize transfer learning strategies in a general setting Zhuang et al. (2020); Weiss et al. (2016); Tan et al. (2018). However, these works do not study transfer learning in the context of VLMs. Recently, Xing et al. Xing et al. (2024a) provides insights to this specific field, but they mainly focus on the adapter-based and the prompt-based transfer learning method which, as we will see in this paper, is not the comprehensive picture of the field. Compared to their work, we provide a comprehensive survey of efficient transfer learning methods used for VLMs. Expanded from prior works, we carefully summarize and classify the transfer learning methods for VLMs into four general categories: **Adapter-based** methods, **Prompt-based** methods, **Model-based** methods, and **Knowledge-based** methods. We also study the corresponding VLMs, datasets, and realistic applications for the VLM transfer learning methods.

The paper is organized as follows: Section 2 introduces a background on vision-language tasks and transfer learning. Section 3 provides insight about pre-trained vision language models, with their architectures and training objectives. Section 4 discusses the taxonomy of efficient transfer learning method for vision-language tasks. Section 5 introduces the real-world application of vision-language transfer learning in various domains. Section 6 discusses the common benchmarks and datasets for vision-language tasks. The paper is concluded in Section 7.

## 2 Background & Foundations

Recent advances in the vision-language models and the transfer learning techniques have significantly expanded the models' understanding of the vision language tasks . In this section, we review prior work on vision–language tasks and discuss how transfer learning has facilitated efficient adaptation across domains and modalities.

### 2.1 Vision Language Tasks

Vision-language tasks involve jointly understanding and reasoning over visual and textual information by enabling cross-modal alignment and comprehension. The early machine learning approaches for handling these tasks relied on manually engineered features and separate models for vision and language processing, such as pre-trained convolutional neural networks (CNNs) for vision and recurrent neural networks (RNNs) for language (Wang et al., 2016; Khamparia et al., 2020; Yu et al., 2017). The outputs of these paths then are integrated to perform the downstream task. Despite obtaining some success, these approaches are limited because of their reliance on task-specific feature engineering and separate processing pipelines, which often struggled to achieve seamless multimodal alignment. Recent advancements in deep learning have enabled the development of unified architectures that use large-scale pre-training to achieve remarkably better performance than the simple combination of CNNs and RNNs (Radford et al., 2021; Chen et al., 2020; Kim et al., 2021; Wang et al., 2021; Li et al., 2021; 2020; Jia et al., 2021; Li et al., 2022b). Building on the foundation laid by unified architectures, more recent large-scale models, which have billion and trillion of parameters, mark a significant step forward in performing vision-language tasks. These recent models leverage massive datasets and transformer-based architectures to process and generate both visual and textual information seamlessly (Achiam et al., 2023; Alayrac et al., 2022; Ramesh et al., 2021; Driess et al., 2023; Huang et al., 2023; Zhang et al., 2023a). For example, GPT-4V (Achiam et al., 2023) exemplifies the trend of incorporating vision into language-dominant models, enabling complex reasoning over image-text inputs via 1.76 trillion parameters. On the side of generative task, DALL · E (Ramesh et al., 2021) boosts test-to-image generation task by synthesizing high-quality images from textual descriptions, showcasing the promising potential of the multimodal generation. Overall, these models not only scale up in terms of parameters and training data but also expand the scope of vision-language field into more complicated and challenging tasks, which brings vision language tasks from experiments to realistic applications.

## 2.2 Transfer Learning

Transfer learning has become a cornerstone technique in machine learning, which enables large models trained on massive datasets to generalize effectively to domain-specific tasks with limited data Socher et al. (2013); Zhuang et al. (2020); Rostami et al. (2022); Zhang et al. (2024c). This paradigm is particularly effective in domains where collecting data is costly or impractical. Early research on transfer learning focused on feature-based approaches, where pretrained models serve as feature extractors, often leveraging architectures such convolutional neural networks (CNNs) pretrained on large image datasets like ImageNet (Deng et al., 2009) for handling vision tasks. Finetuning has since emerged as a flexible and powerful approach, allowing the adaptation of pretrained models to target tasks by updating their parameters in a partial or full manner.

Transfer learning has diversified into several sub-domains, each addressing specific challenges, including: (i) **Domain Adaptation**, which focuses on transferring knowledge from a source domain with labeled data to a target domain with different distribution in which we only have unlabeled data. Techniques in domain adaptation often involve aligning feature spaces across domains. (ii) **Zero/Few-Shot Learning**, which aims to generalize the knowledge from seen classes to unseen classes or tasks without access or with very limited access to labeled training data for those classes. (iii) **Continual Learning**, which addresses the challenge of learning sequentially from a stream of tasks without forgetting previously learned tasks. (iv) **Multi-Task Learning**, which focuses on training models on multiple tasks simultaneously to encourage shared representations that help improve the overall tasks' performance. Transfer learning has evolved from simple feature reuse to sophisticated frameworks that span a diverse set of applications and challenges. Its sub-domains continue to address critical gaps for the case vision-language tasks, enabling models to perform robustly in dynamic and resource-constrained environments. Our goal is to survey transfer learning methods that are designed to be used on VLMs for handling vision-language tasks more efficiently.

As vision-and-language tasks continue to evolve, transfer learning has played a crucial role in improving model generalization across diverse datasets and applications. Early approaches relied on task-specific models trained from scratch or finetuned on relatively small datasets. However, with the rise of large-scale vision-language datasets and the success of self-supervised learning, a shift toward pre-trained large Vision-Language Models (VLMs) has emerged. These models, trained on large-scale multimodal corpora, serve as powerful foundations that can be adapted to downstream tasks with minimal finetuning. In the next section, we explore the architectures and training objectives of these pre-trained Large VLMs, highlighting how they enable transfer learning at scale and achieve state-of-the-art performance across vision-language benchmarks.

# 3 Pre-trained Large Vision Language Models

In recent years, pre-trained large Vision-Language Models have emerged as a powerful class of AI models that bridge the gap between computer vision and natural language processing. These models are trained on massive datasets containing both images and text, enabling them to understand and generate complex visual and textual content. Pre-trained large vision-language models are the fundamental modern tools to solve vision-language tasks and are the backbones that transfer learning methods in the area built on (Kim et al., 2021; Radford et al., 2021). In recent works, Du et al. (2022); Zhang et al. (2024b); Wu et al. (2023a); Liang et al. (2024) provide detailed summarization about pre-trained large vision language models. In our paper, to offer a more comprehensive insight about VLMs and help understand the transfer learning methods in this area, we survey the most popular VLM architectures in section 3.1 and the objective functions that are used for training them in section 3.2. Additionally, we present the most popular large-scale pre-trained VLMs in temporal order in Figure 1.

## 3.1 VLM Architectures

Various architectures have been designed to bridge the gap between visual and textual modalities (See Figure 2). A general approach to categorize VLM architecture is based on how the vision and text features are integrated, namely, **Contrastive Dual Encoder** architecture, **Encoder-Decoder** architecture and **Multi-modal LLM** architecture. We briefly survey these categories.

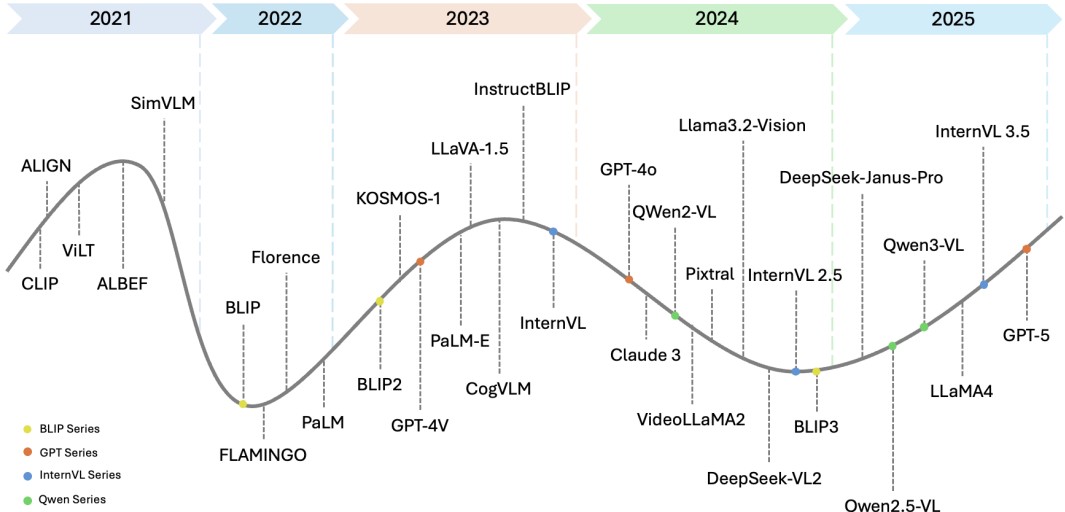

Figure 1: Development of large-scale pretrained vision language model

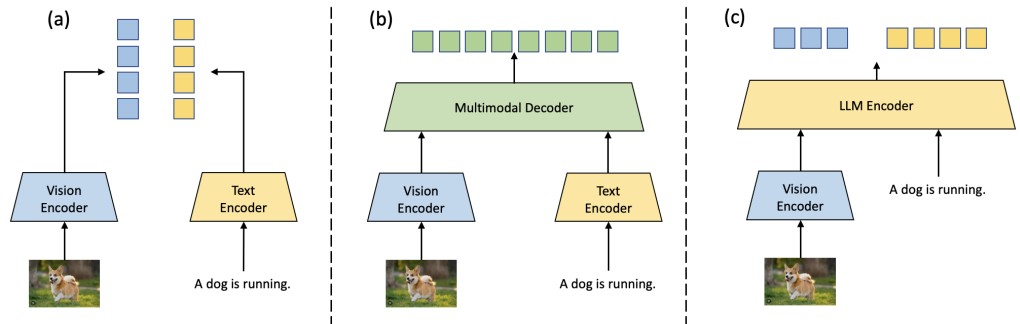

Figure 2: Diagram shows the three different architecture of pretrained large vision language model. **(a)** Contrastive Dual Encoder Architecture, **(b)** Encoder-Decoder Architecture, **(c)** Multi-modal LLM Architecture.

### 3.1.1 Contrastive Dual Encoder Architecture

Some vision-language models utilize independent vision and text encoders to generate features from different modalities and align them into the shared embedding space (see Figure 2 (a)). During pretraining, the model is optimized using a contrastive loss, such as InfoNCE loss, that encourages matched image–text pairs to have similar embeddings, while pushing apart mismatched pairs.

For example, Radford et al. (2021) use vision transformer (Dosovitskiy et al., 2021) as vision encoder and the BERT model (Kenton & Toutanova, 2019) as the text encoder. The encoders are further trained via a large-scale image-pair dataset with contrastive loss to guarantee the alignment of feature from both encoders for matching image and text features. Jia et al. (2021) adopt a similar architecture and train the model on a more general and noisy dataset to achieve more advanced performance. Yuan et al. (2021), on the other hand, expand the feature representation from scene-level to object level and enable the video and depth image retrieval. However, although the contrastive dual encoder architecture models are computationally efficient and well-suited for retrieval-style tasks (e.g., image–text retrieval or zero-shot classification), they generally lack fine-grained cross-modal reasoning since the encoders operate independently.

### 3.1.2 Encoder-Decoder Architecture

Different from contrastive dual encoder architecture models, the encoder–decoder architecture models integrate visual and textual information within a single multimodal transformer framework(see Figure 2 (c)). Both image and text inputs are first tokenized into unified embeddings, with a vision encoder producing patch tokens and a text tokenizer producing word tokens, which are then jointly processed by a shared transformer. This allows bidirectional cross-modal attention at every layer, enabling deep fusion and reasoning between modalities.

For example, Kim et al. (2021) use transformer blocks as multimodal decoder. The input image and text are first converted to feature tokens via the modality-specific encoder. The image feature and the text feature are concatenated together, after applying the modality-specific type embedding and the positional embedding, to form the input of the multimodal decoder. Li et al. (2021) take a further step such that they apply the image-text contrastive loss to the unimodal features before the fusion. During the pre-training, the model learns to align the image feature and the text feature from the same image-text pair, via the contrastive loss, before fusion. The image-text matching loss and the masked-language-modeling loss are then applied to learn the multimodal interaction between the image and the text.

However, although encoder-decoder architecture models are more powerful solving the reasoning task than the contrastive dual encoder architecture models, the multimodal decoders often smaller or specially-designed multimodal transformers, rather than very large language models (LLMs) with hundreds of billions of parameters. This limits their pure language reasoning capacity. On the other hand, as the current large-scale LLMs are naturally capable of complicated reasoning tasks, building a relatively small multi-modal decoder module on the top of the LLM may harm its performance on reasoning-centric tasks.

### 3.1.3 Multi-Modal LLM Architecture

To fully utilize the reasoning capacity of the SOTA large language models, multi-modal LLM architecture models aims to integrate the image features into the LLMs to enable multi-modal reasoning. In this design, image features extracted by a powerful vision backbone are projected into the LLM's token space and fused via cross-attention or lightweight connector networks. This enables the model to perceive and reason jointly across modalities while leveraging the extensive world knowledge, linguistic fluency, and reasoning capability of the underlying LLM. Compared with traditional encoder–decoder or unified multimodal architectures, multimodal LLMs offer superior scalability, modularity, and instruction-following ability, allowing the reuse of existing text-only LLMs and rapid adaptation to multimodal tasks through instruction tuning.

For example, LLaVA (Liu et al., 2023) adopts CLIP's ViT as the visual encoder and the pre-trained Vicuna model as the LLM backbone. The image features from the visual encoder are projected, by a trainable projection matrix, to the LLM backbone's embedding space and become part of the input to the LLM. BLIP-2 (Li et al., 2023) model, on the other hand, finetune a light-weight querying transformer (Q-Former) as the vision-language project to adopt the image feature into the LLM. BLIP-2 freezes the visual encoder and the LLM during the finetuning and only update the Q-Former, enabling a generic and efficient modality interaction. Flamingo (Alayrac et al., 2022) applies novel perceiver resampler as the vision-language projector and inserts *gated xattn-dense layers* in the LLM layers to align image feature into the text encoder.

After discussing the three major architectural paradigms of the VLMs, we now turn to the training objectives that guide how these models learn cross-modal alignment and reasoning. While architectures define the interaction between vision and language, training objectives determine the nature of this interaction—ranging from the contrastive alignment in dual encoders to the matching and generative learning in multimodal LLMs.

### 3.2 VLM Training Objective Functions

The training objective functions for VLMs play a pivotal role in enabling effective cross-modal alignment and understanding. We summarize the commonly used objective functions into three categories, including contrastive objectives, generative objectives and matching objectives.

**Contrastive objectives** aim to align the representations of different modalities within the same pair closer to each other and to be distant from the representations of the other pairs. The most common contrastive loss used for training VLMs is based on the InfoNCE loss (Radford et al., 2021; Jia et al., 2021; Chen et al., 2024b) which aims to maximize the alignment of features for positive pairs while minimizing the similarity to the features of negative pairs. The image to text loss is expressed as:

$$\mathcal{L}_{\text{I2T}} = -\frac{1}{N} \sum_{i=1}^{N} \log \frac{\exp(\text{sim}(f_I(x_i), f_T(y_i))/\tau)}{\sum_{j=1}^{N} \exp(\text{sim}(f_I(x_i), f_T(y_j))/\tau)}, \tag{1}$$

while the text to image loss is:

$$\mathcal{L}_{\text{T2I}} = -\frac{1}{N} \sum_{i=1}^{N} \log \frac{\exp(\text{sim}(f_T(y_i), f_I(x_i))/\tau)}{\sum_{j=1}^{N} \exp(\text{sim}(f_T(y_i), f_I(x_j))/\tau)} \tag{2}$$

By combining the two, the final contrastive loss is formed as below:

$$\mathcal{L}_{\text{contrastive}} = \mathcal{L}_{\text{I2T}} + \mathcal{L}_{\text{T2I}}, \tag{3}$$

where $f_I(x_i)$ represents the embedding of image $x_i$, $f_T(y_i)$ represents the embedding of text $y_i$, and $\tau$ is the temperature hyper-parameter to control the sharpness of the softmax distribution.

**Generative objectives** generative objectives for image modality are easier to define because visual data has a clear, structured pixel-based representation that allows straightforward metrics like pixel-wise differences Goodfellow et al. (2014); Shin et al. (2017); Rostami et al. (2019). In addition to bi-modality, the challenge for defining generative objectives for VLMs is that language data is not as structured as visual data. let the model learn the semantic property of features to generate the corresponding image or text for either prediction or filling a missing part (Chowdhery et al., 2023; Driess et al., 2023; Achiam et al., 2023). Masked language modeling (MLM) is a common approach to learn the probability distribution of the sequences of words or tokens in a given context by measuring how well a model can predict the word in a sequence given the preceding or surrounding context. The objective function is denoted as:

$$\mathcal{L}_{\text{LM}} = -\frac{1}{T} \sum_{t=1}^{T} \log P(w_t \mid w_1, w_2, \ldots, w_{t-1}), \tag{4}$$

where $w_t$ is the ground truth next token at position $t$.

On the other hand, masked image modeling (MIM) (Li et al., 2022b; Wang et al., 2021) is used to learn the image representation by predicting the masked part of the image patch as:

$$\mathcal{L}_{\text{MIM}} = -\sum_{i \in M} \log P(v_M \mid v_U), \tag{5}$$

where $v_M$ is the masked token and $v_U$ represents the unmasked tokens.

**Matching objectives** help understand the relationship between images and their corresponding text descriptions. The goal of image-text matching is to determine whether a given image and a piece of text (e.g., a caption or description) are semantically aligned or not. Different from contrastive objective, matching objectives are generally a type of binary classification loss instead of measuring similarity as:

$$\mathcal{L}_{\text{ITM}} = -\frac{1}{N} \sum_{i=1}^{N} \left[ y_i \log(\hat{y}_i) + (1 - y_i) \log(1 - \hat{y}_i) \right], \tag{6}$$

where $y_i$ is the ground truth that 1 means match and 0 for otherwise, and $\hat{y}_i$ is the predicted probability that whether the paired image and text matches with each other.

As pre-trained Large VLMs continue to advance, their widespread adoption has highlighted both their strengths and challenges. While these models achieve state-of-the-art performance across various vision-language tasks, the massive scale introduces significant computational and deployment costs, making finetuning or adapting them for specific applications impractical in many scenarios. To address these limitations, efficient transfer learning methods have been developed to leverage the rich knowledge encoded in large VLMs while reducing resource consumption. In the next section, we explore different methods which enable effective model adaptation with minimal compute- and temporal- cost, making large VLMs more accessible and scalable for real-world applications.

## 4  Transfer Learning Approaches for Vision-Language Tasks

Transfer learning aims to enable large models, which are trained on massive datasets, to generalize effectively to domain-specific tasks with limited data. In this section, we categorize the efficient transfer learning approaches for VLMs into four categories, namely **Adapter Based Method** in section 4.1 , **Prompt Based Method** in section 4.2, **Knowledge Based Method** in section 4.4 and **Pipeline Based Method** in section 4.3. We summarize methods in each category.

### 4.1  Adapter-Based Methods

End-to-end finetuning is inefficient for large models and often would lead the model to become overfit on the finetuning datasets. Adapter-based transfer learning methods introduce trainable small modules during the finetuning stage to enable efficient adoption of pre-trained models to new tasks without updating all the parameters Houlsby et al. (2019); Jin et al. (2021b); Sung et al. (2022). Houlsby et al. (2019) insert small trainable feedforward networks modules in every transformer blocks, and consider those modcules to be **adapters**. Since only a small number of parameters are updated during finetuning, adapter-based methods significantly reduce the memory requirement for model storage and computational cost when finetuning a pretrained VLM on several downstream tasks. When dealing with VLMs, adapters are not constrained to layer-wise FFNs only. Among all the vision-language adapter-based transfer learning methods, we further divide them into **layer-wise** adapters, which are repeatedly inserted in layers of the original model, and **module-wise** adapters, which is the independent modules that only plugged into the VLM once.

#### 4.1.1  Layer-Wise Adapter

The Layer-wise adapters are often inserted into transformer layers, shown in Figure3(a). The adapter layers receive the output from the preceding transformer blocks as their input and generate a light-weight transformation. The transformed features are then passed into the next transformer layer.

Li & Sun (2023) propose LiFT-Adapter which consists of two layers: bottleneck one fully-connected layer and one non-linear activation function. The adapter modules are inserted after the MLP layers in the transformer blocks of the vision encoder while the vision encoder remains frozen. Yang et al. (2022a), similarly, apply adapters on both of the frozen pre-trained vision model and the language model. As the backbones are frozen, only the adapters and normalization layers for each transformer block are updated during the finetuning stage, enabling both mitigating catastrophic forgetting and modality alignment. While the above two methods apply regular static adapter to the base-model's architecture, other methods propose more different designs of adapter. Chen et al. (2025a) introduce a shared-weight multi-modal spatio-temporal adapter (MSTA) for video-text matching. MSTA introduces a spatio-temporal description-guided consistency constraint such that in the vision encoder branch, the output of adapter goes to both the spatial up-sampling layer and the temporal up-sampling layer to mitigate over-fitting and enhance generalizability. Zhang et al. (2023b) and Gao et al. (2023) introduce llama-adapter, for LLaMA (Touvron et al., 2023) based models, with adaptation prompt and gated zero-init attention to efficiently finetune the LLaMA model for a variety of downstream tasks. Yu et al. (2024) introduce MoE-Adapters for continual learning. MoE-Adapters contain multiple adapters, inserted after the multi-head attention block of each transformer layer in both the visual encoder and the text encoder. The adapters are selected by the task-specific routers for different tasks. Wu et al. (2024), on the other hand, propose Dynamic Architecture Skipping by utilizing adapters to replace parts of the vision-language layers. The method observes the significance of each module in a reinforcement

| Category | Sub-Category | Method | Transfer Learning Domain | Downstream Task |
|---|---|---|---|---|
| Layer-Wise | - | LiFT-Adapter (Li & Sun, 2023) | Few Shot Learning, Novel Class Discovery | Image Classification |
| | | LLaMA-Adapter (Zhang et al., 2023b) | Efficient Finetuning | Multi Modal Reasoning |
| | | LLaMA-Adapter v2 (Gao et al., 2023) | Efficient Finetuning | Multi Modal Reasoning |
| | | MoE-Adapter (Yu et al., 2024) | Continual Learning | Image Classification |
| | | CLIP-LoRA (Zanella & Ben Ayed, 2024) | Few Shot Learning | Image Classification |
| | | DAS (Wu et al., 2024) | Efficient Finetuning | VQA, NLVR, Retrieval |
| | | BiLM (Yang et al., 2022a) | Zero Shot Learning | Video Question Answering |
| | | MSTA (Chen et al., 2025a) | Efficient Finetuning | Video Text Retreival |
| | | MMA (Yang et al., 2024) | Domain Generalization | Image Classification |
| Module-Wise | Cross-Modal | Tip-Adapter (Zhang et al., 2021) | Few Shot Learning, Efficient Finetuning | Image Classification |
| | | CLIP-Adapter Gao et al. (2024) | Few Shot Learning, Efficient Finetuning | Image Classification |
| | | TDA (Karmanov et al., 2024) | Domain Adaptation | Image Classification |
| | | Meta-Adapter (Song et al., 2023) | Few Shot Learning | Image Classification |
| | | Prompt-Aware Adapter (Zhang et al., 2025) | Efficient Finetuning | Visual Question Answering |
| | | Cross-Modal Adapter (Jiang et al., 2025) | Efficient Finetuning | Image Text Retrieval |
| | Uni-Modal | SgVA-CLIP (Peng et al., 2023) | Few Shot Learning | Image Classification |
| | | SVL-Adapter (Pantazis et al., 2022) | Few Shot Learning | Image Classification |
| | | RAIL (Xu et al., 2024) | Continual Learning | Image Classification |

Table 1: Summary of adapter-based methods

learning manner, and the less significant module is skipped, or replaced by adapters to avoid drastic change in hidden features from previous layers. Yang et al. (2024) propose a multi-modal adapter (MMA) for Clip-based few shot learning. MMA connects the frozen higher-layers of text encoder and visual encoder through a shared projection layer and makes features more discriminative and generalizable.

Moreover, other than providing novel designs such as those mentioned above, Zanella & Ben Ayed (2024) offer analysis of applying LoRA Hu et al. (2021) to enable few-shot learning with VLMs to explore the application of LoRA-like adapters in vision-language transfer learning. Similarly, Sung et al. (2022) analyze the performance of different parameter-efficient training techniques, such as adapter, hyperformer and compactor, for image-text and video-text tasks. The methods are summarized in Table 1 layer-wise section.

### 4.1.2 Module-Wise Adapter

Different from layer-wise adapters which are repeatedly added to the original model, module-wise adapters take a different strategy through designing independent modules outside of transformer layers Song et al. (2023); Cai & Rostami (2024a); Gao et al. (2024); Zhang et al. (2021). The designed module is usually used only once. The adapter is then plugged in between the original backbone's modules, e.g., between the visual encoder and the multimodal encoder. Compared to the layer-wise adapters, module-wise adapters aim for high-level feature alignment and are more modular and flexible. As adapters are independent and used only once, they increase the scalability for large models. Within module-wise adapters, they can be further categorized into cross-modal adapters and uni-modal adapters.

**Cross-Modal Adapter** Cross-modal adapters take both image and text modalities as inputs, either by a single adapter layer or separate adapter layers. The general architecture is shown in Figure3(b). Jiang et al. (2025) introduce a cross-modal adapter, it enables encoder-level cross-modal interactions by sharing adapters' weights between two modalities rather than introducing explicit feature interactions. Such a scheme allows an implicit cross-modal interaction, which will facilitate the re-alignment of vision and language feature spaces for the cross-modal retrieval task. Song et al. (2023) propose a single cross-modal adapter layer for aligning inputs from vision and text encoders using the attention-based adapter to refine the CLIP features, guided by a few samples in an online manner. The adapter layer takes queries from the support images and Key/Query from text labels to refine the category embedding for the target image. On the other hand, Clip-Adapter (Gao et al., 2024) adopts two separate adapters, one for each modality, which consist of small bottleneck linear layers, $Av(\cdot)$ and $At(\cdot)$, to handle image features and text features. Inspired by Clip-Adapter, Zhang et al. (2021) adopt the idea of multi-layer perceptron and residual connection, but also propose a novel cache-based model to obtain the weights from few-shot visual features and ground truth labels. Following the idea of cache-based method, Training-free Dynamic Adapter (TDA) (Karmanov et al., 2024) expands the design of cache models. It constructs and updates two key-value caches to store the knowledge of a stream of test samples, and uses the two caches to generate positive and negative predictions which are combined with the CLIP predictions to produce the final prediction. Zhang et al. (2025) propose

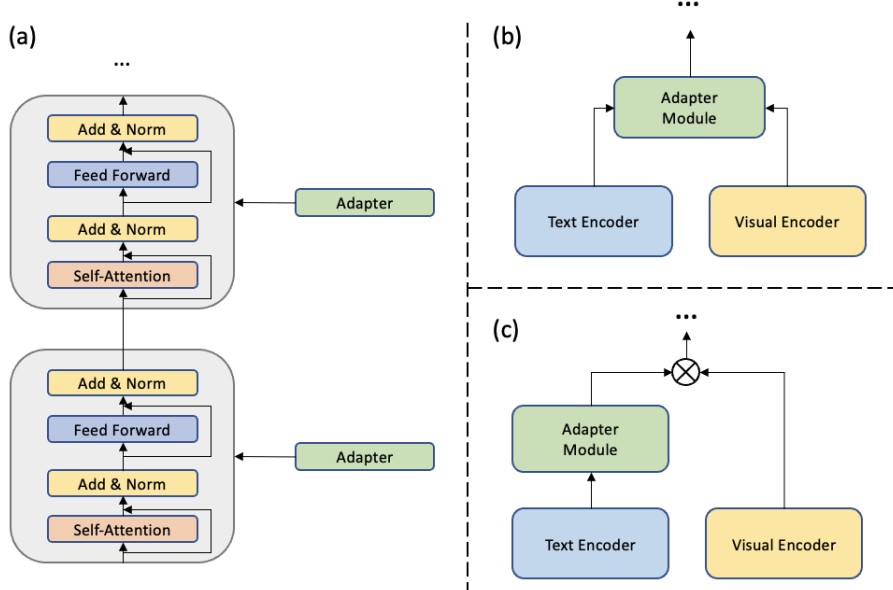

Figure 3: Taxonomy of Adapter-based Method. **(a):** layer-wise adapter, **(b):** cross-modal adapter, **(c):** uni-modal adapter.

a novel prompt-aware adapter to align the image and the text input. It is designed to dynamically embed the visual inputs based on the specific focus of the text prompt, by using both global and local textual features to capture the most relevant visual clues from the prompt at both coarse and fine granularity levels. The methods are summarized in Table1 cross-modal section. TAM-CL Cai et al. enables continual learning through a task-attentive layer which receives its input form the cross-modal transfer layers and adapt the model to generalize on a sequence of continually observed tasks.

**Uni-Modal Adapter**   Different from cross-modal adapters, uni-modal adapters take one modality as input to boost the generalizablity of backbone model in transfer learning scenarios. The diagram is shown in Figure3(c). Existing uni-modal adapters focus on the enhancement of visual features. Xu et al. (2024) design regression-based analytic incremental learning (RAIL) which utilizes a recursive ridge regression-based visual adapter to learn from a sequence of domains in a non-forgetting manner and decouple the cross-domain correlations by projecting features to a higher-dimensional space. Similarly, Pantazis et al. (2022) propose a vision-only adapter that takes input from an additional CLIP visual encoder. The adapted visual features are used to fuse with the output at classification level to guide zero-shot and low-shot image classification. Moreover, Peng et al. (2023) adopt a two-layer perceptron as vision adapters to support visual-specific contrastive loss between query images and support images in few-shot learning setting. By adopting visual adapters, visual features can be better aligned with the text features extracted by LLMs, which in turn improve the overall performance of VLMs. The methods are summarized in Table 1 uni-modal section.

## 4.2   Prompt-Based Methods

Prompt-based methods enable transfer learning of VLMs using additional image or text features and trainable variables to guide the VLM to perform the downstream task without extensive finetuning Ge et al. (2023); Jia et al. (2022); Qian et al. (2023); Cai & Rostami (2024b) . By freezing the backbone model, prompt-based methods adopt the original knowledge from pretrained VLMs to the downstream task with the guidance of well-designed additional inputs and thus make these methods parameter-efficient and flexible. In this section, we survey textual prompts in Sec 4.2.1, visual prompts in Sec 4.2.2 and multimodal prompts in Sec 4.2.3.

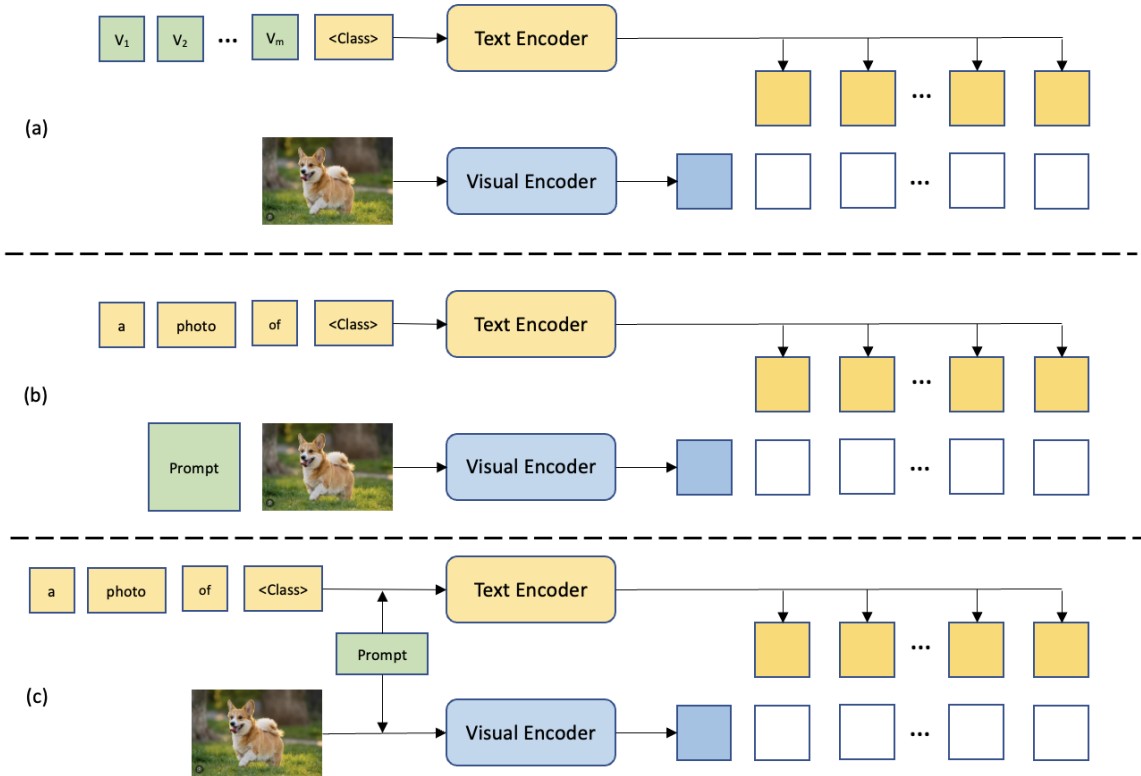

Figure 4: Taxonomy of Prompt-based Methods. **(a):** Textual Prompt, **(b):** Visual Prompt, **(c):** Multimodal Prompt.

| Category | Method | Transfer Learning Domain | Downstream Task |
|---|---|---|---|
| Textual Prompt | PLOT (Chen et al., 2022a) | Few Shot Learning | Image Classification |
| | FEWVLM (Jin et al., 2021a) | Zero/Few Shot Learning | Image Classification, Image Captioning, VQA |
| | CoCoOp (Zhou et al., 2022a) | Domain Generalization | Image Classification |
| | UDA (Ge et al., 2023) | Domain Adaptation | Image Classification |
| | CoOp (Zhou et al., 2022a) | Few Shot Learning | Image Classification |
| | MPA (Chen et al., 2023) | Unsupervised Domain Adaptation | Image Classification |
| | PODA (Fahes et al., 2023) | Zero Shot Learning, Domain Adaptation | Semantic Segmentation |
| | ProDA (Lu et al., 2022) | Few Shot Learning | Image Classification, Object Detection |
| | ProGrad (Zhu et al., 2023) | Few Shot Learning, Domain Generalization | Image Classification |
| | StyleGAN-NADA (Gal et al., 2022) | Domain Generalization | Image Generation |
| | TPT (Shu et al., 2022) | Zero Shot Learning, Domain Adaptation | Image Classification |
| | TriMPL (Liu et al., 2024c) | Domain generalization, Few Shot Learning | Image Classification |
| | PMP (Liu et al., 2025) | Domain Generalization, Few Shot Learning | Image Classification |
| Visual Prompt | Visual Prompt (Bahng et al., 2022) | Efficient Fine Tuning | Image Classification |
| | SHIP (Wang et al., 2023d) | Zero Shot Learning | Image Classification |
| | RePrompt (Rong et al., 2023) | Few Shot Learning, Domain Generalization | Image Classification |
| | VPT (Jia et al., 2022) | Efficient Fine Tuning | Image Classification |
| | VL-PTM (Wen et al., 2022) | Few Shot Learning | Text Classification |
| | CPT (Yao et al., 2024) | Few/Zero Shot Learning | Visual Relation Detection, Visual Reasoning, VQA |
| Multimodal Prompt | APT (Wu et al., 2023b) | Efficient Fine Tuning | Text-Image Generation, Image Classification |
| | TRIPLET (Qian et al., 2023) | Continual Learning | Visual Question Answering |
| | MaPLe (Khattak et al., 2023) | Domain Generalization | Image Classification |
| | MVLPT (Shen et al., 2024) | Few Shot Learning, Domain Generalization | Image Classification |
| | PADCLIP (Lai et al., 2023) | Unsupervised Domain Adaptation | Image Classification |
| | PMHANet (Liu et al., 2022) | Efficient Fine Tuning | Image Classification |
| | UPL (Huang et al., 2022) | Efficient Fine Tuning | Image Classification |
| | ViTiS (Engin & Avrithis, 2023) | Zero/Few Shot Learning | Visual Question Answering |
| | HiCroPL (Zheng et al., 2025) | Efficient Fine Tuning | Image Classification |

Table 2: Prompt-based methods

### 4.2.1 Textual Prompt

Textual prompts, presented in Figure4(a), are the natural language input or the add-on parameters to the textual inputs or features which aim to guide the model's understanding of the input task by enriching the semantic knowledge of the given input text. It has been observed that enriching input text can help understanding the vision modality better and can improve generalization Shangguan et al. (2025b).

The most intuitive textual prompt is based on the human-readable natural language. Ge et al. (2023) introduce text-based DAPrompt for image domain adaption tasks, which aims to embed domain information into domain-specific prompts, which is a form of representation generated from natural language, and then align the image features of the same domain and push them distant from the ones of other domains. Fahes et al. (2023) propose PODA for zero-shot image domain adaptation using prompts. PODA uses the natural language description of target image to match the source image via CLIP-based model, and bring the source image features in CLIP feature space closer to its imaginary counterpart in the target domain. Besides using the natural language as the textual prompt, other methods rely on learnable context parameter to enrich the textual features. Zhou et al. (2022b) propose Context Optimization (CoOp) such that instead of adopting a context such as "a photo of" or any other phrase for CLIP text input, the method turns the context into learnable vectors. By freezing the text encoder and the image encoder, the learnable contexts are updated through the back-propagation and are made to be efficient for few-shot learning. Following CoOp, Conditional Context Optimization (CoCoOp) (Zhou et al., 2022a) extends the idea by further learning a light weight neural network that generates an input-conditional token for each image, combined with the learnable context vector. Zhu et al. (2023) develope another variant of CoOp by training learnable context with gradient regularization strategy. The method first measures the gradient *general direction*, which is the KL loss between the zero-shot original CLIP and the few-shot finetuned model, and then decomposes the gradient into components which are orthogonal to the *general direction* and the gradient parallel to the *general direction*. The prompts are then updated accordingly. Similarly, Shu et al. (2022) provide Test-time Prompt Tuning (TPT). TPT tunes the prompts, which are learnable context features, on the fly using only the test sample. The tuned prompt is adapted to each task, making it suitable for zero-shot generalization without requiring any task-specific training data or annotations.

While the above methods adopt the single prompt, other methods utilize the combination of multiple prompts. Chen et al. (2022a) introduce multiple learnable context vectors as the prompt and use optimal transport as the metric to align the prompt features with the local features of an image, achieving the alignment of different modalities in a more fine-grained and comprehensive way. Similarly, ProDA (Lu et al., 2022) generates multiple learnable prompts. By integrating the text with all the prompts, ProDA forms a weight distribution of the output embedding features for each object class in the feature space. The image features are then aligned with the best-matched distribution. Chen et al. (2023) provide a multi-prompt method for unsupervised domain adaptation. For source domain and target domain, they propose both domain-invariant features and domain-specific features and align them with the input text to learn domain-shared and domain-specific knowledge. FEWVLM (Jin et al., 2021a) provide a prompt-based method for low resource learning via analyzing the performance of different vision-lamguage tasks on zero/few shot learning with different prompts. On the other hand, Liu et al. (2025) propose a progressive implementation of multi-prompt (PMP). PMP introduces multiple prompts in a step-by-step manner to focus on various information and utilizes a late attaching mechanism to defer the interactions of prompts and features to a deeper encoding layer to reduce the overfitting. All the methods are summarized in Table 2 textual prompt section.

### 4.2.2 Visual Prompt

Visual prompts refer to the prompts attached to the visual inputs to augment the visual semantics. While textual prompts can be crafted manually, visual prompts often require complex transformations that need to be learned or optimized. The diagram of the visual prompt is presented in Figure 4(b).

Similar to the textual prompt based method, some visual prompt based methods adopt trainable variables on the vision side as visual prompts. Jia et al. (2022) propose Visual Prompt Tuning (VPT) for downstream recognition tasks. VPT prepends learnable prompt parameters to each visual encoder transformer layer's input while keeping the whole model frozen. Bahng et al. (2022) propose a trainable visual prompt for input image. The image feature is summed with the visual prompt to form a prompted image, which is then goes through the frozen visual encoder.

Other than directly applying trainable parameters to the visual input, some methods explore generating pseudo images as visual prompts. Wen et al. (2022) freeze CLIP and use the class name to generate pseudo images. The pseudo image features are combined with the class name to match with the text content

through CLIP-style contrastive loss. Wang et al. (2023d) introduce Synthesized Prompt (SHIP) that trains a generator model to synthesize image features. The image features from the visual encoder are sent to a variational autoencoder (VAE) and are confined to a prior distribution. The sample from the distribution is then sent to the text encoder, prepending to the class name, for feature reconstruction.

Different from methods above, some methods apply existing images as the visual prompts. Rong et al. (2023) propose retrieval-enhanced visual prompt learning (RePrompt) for few-shot classification with additional realistic images. RePrompt constructs a retrieval database from either training examples or external data, if available, and uses a retrieval mechanism to enhance multiple stages of a simple visual prompt learning baseline, thus narrowing the domain gap. Yao et al. (2024) propose the colorful prompt tuning (CPT) which reformulates visual grounding into a fill-in-the-blank problem with the color-based co-referential markers in image and text. CPT masks the proposed region with the natural visual marker of a different color as visual sub-prompt, and rewrite the text to select the color of bounding box of the targeted object as text sub-prompt. Hence, the visual grounding can be reformulated into a simple fill-in-the-blank problem. All the visual prompt methods are summarized in Table2 visual prompt section.

### 4.2.3 Multi-modal Prompt

Unlike uni-modal prompts that rely solely on either vision or language, multi-modal prompts provide a cohesive mechanism to influence how models process and align information across both modalities. The architecture of multi-modal prompt is shown in Figure 4(c).

An intuitive approach to generate the multi-modal prompt is to treat the prompt from both modalities separately, such that the prompt of each modalities remains relatively independent and balanced. Qian et al. (2023) introduce a multimodal prompt for VLMs with multi-modal encoders such as ALBEF (Li et al., 2021) for the continual learning of VQA tasks. By freezing the backbone, they train the visual prompt for image input, the text prompt for text input, and the fusion prompt for multi-modal encoder input. The fusion prompt is made by the weighted interaction of the two single-modality prompts, where each prompt is divided into the task-agnostic general prompt $G$ and the task-specific expert prompt $E$. Similarly, Cai & Rostami (2024b) propose CluMo, which is also adopted for continual learning of VQA tasks using VLMs with a multi-modal encoder. Instead of adopting uni-modal prompt for fusion prompt, CluMo designs uni-modal prompt keys, which are trained via $K$-means clustering on the training data. The visual prompt key and the text prompt key are then combined to determine the best matched fusion prompt which is used for the multi-modal encoder. Zang et al. (2022) propose the Unified Prompt Tuning (UPT), which learns a tiny neural network to jointly optimize the prompts across different modalities. UPT designs a layer-wise unified modal-agnostic prompt that can be split into the corresponded text prompt and image prompt, which are separately input to the text encoder and the image encoder.

Other than treating the different modalities' prompt relatively independently, some methods rely on generating the prompt of one modality conditioned on the other modality's prompts and features. For example, Huang et al. (2022) provide a novel design of unsupervised learning in multi-modal prompt setting for unlabeled datasets. While the text prompt is shared learnable prompt, the method applies CLIP-style encoding for both the image and the text inputs to generate pseudo-labels for unlabeled images, which serve as the visual prompt for the visual input. Khattak et al. (2023) propose Multi-modal Prompt Learning (MaPLe) which attaches the text prompt to the text input, attaches the visual prompt to the image input. The image prompt is conditioned on the text prompt via a vision-language coupling function, which is trained along with the prompt. Similarly, Engin & Avrithis (2023) propose ViTiS for few-shot video question answering. It trains a visual mapping network via a LLM so that the visual prompts are transformed, conditioned on the text, into the textual feature space and input to the frozen language model. ViTiS also proposes a novel attention-level text prompt such that the prompts are used as inputs to every attention layer. Zheng et al. (2025), on the other hand, introduce a Hierarchical Cross-modal Prompt Learning (HiCroPL) framework that establishes bidirectional knowledge flow between text and vision modalities. It adopts a layer-wise modality fusion strategy such that in early layers, text prompts inject relatively clear semantics into visual prompts through a hierarchical knowledge mapper, enhancing the representation of low-level visual semantics. In later layers, visual prompts encoding specific task-relevant objects flow back to refine text prompts, enabling deeper alignment. All the methods are summarized in Table 2 multimodal prompt section.

| Category | Method | Transfer Learning Domain | Downstream Task |
|---|---|---|---|
| | CLAP (Jha et al., 2024) | Continual Learning | Image Classification |
| | VT-CLIP (Qiu et al., 2021) | Few Shot Learning | Image Classification |
| | VL-Few (Ma et al., 2024) | Few Shot Learning | Visual Question Answering |
| | -(Chen et al., 2022b) | Domain Adaptation | Visual Understanding |
| | MAPL (Mañas et al., 2022) | Few Shot Learning | VQA, Image Captioning |
| Cross-Model Interaction | CDCIN (Zhang et al., 2024a) | Few Shot Learning | Visual Question Answering |
| | CIRPLANT (Liu et al., 2021) | Domain Adaptation | Image Retrieval |
| | LingoCL (Ni et al., 2024) | Continual Learning | Image Classification |
| | CaFo (Zhang et al., 2023c) | Few Shot Learning | Image Classification |
| | RanPAC (McDonnell et al., 2024) | Continual Learning | Image Classification |
| | TaskRes (Yu et al., 2023) | Domain Generalization | Image Classification |

Table 3: Summary of pipeline-based methods

## 4.3 Model-Based Methods

In the subsections above, we discussed the adapter-based methods and the prompt-based methods, and both of them introduce extra learnable parameters to boost the model's performance on downstream tasks. Besides inserting additional parameters, the model-based methods focus on the modification of the model's architecture itself and come up with the novel design of the model's training pipeline, loss objective and architecture. In this section, we survey and categorize these methods into cross-modal pipeline and novel-architecture pipeline methods.

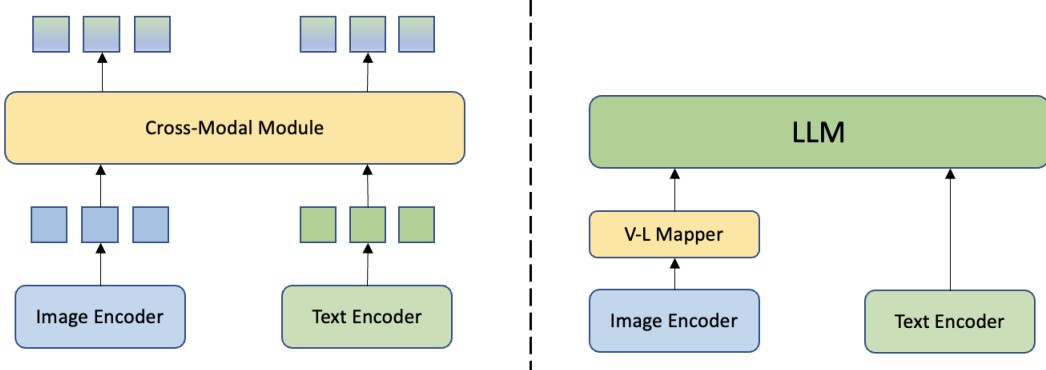

Figure 5: Diagram for different model-based cross-modality pipelines. **(left)**: Methods that use features from one modality to guide the features of another modality. **(right)**: Methods that map the visual features into the textual space via LLM.

A group of methods apply cross-modality pipelines by using features from one modality to guide the features of another modality. Jha et al. (2024) propose probabilistic modeling for downstream tasks. Using the designed visual-guided attention inference module, the proposed method learns the functional space of task-specific posterior distributions based on text features that are aligned with their visual counterpart. Similarly, Qiu et al. (2021) propose visual-guided attention module that takes text features as query and image features as key and value. The module contains co-attention layers that helps explore the informative regions of image that is related to the text of different category, and associates them with more attention weights. While the above two methods use visual-guided textual features, Zhang et al. (2024a) propose a textual-guided visual feature method, named *visual information entropy*, which represents the spatial distribution of visual features guided by questions. To minimize visual information entropy, a multi-modal feature adaptive module is designed to improve cross-modal interaction which calculates the visual information entropy before and after interaction. The novel information consistency loss is then used to minimize the difference between the visual information entropy of the two stages.

Another approach to ensure cross-modal interaction is to map the visual features into the textual space via LLMs. Ma et al. (2024) explicitly extract the cross-modal token through a vision mapper and a vision-language mapper. After the encoder, the vision feature is sent to the vision mapper to extract vision representation alignment token, while vision features and text features are sent to the vision language

| Category | Method | Transfer Learning Domain | Downstream Task |
|---|---|---|---|
| Knowledge-Based | Selective Decomposition (Khan et al., 2024) | Zero Shot Learning | Visual Question Answering |
| | KGENVQA (Cao & Jiang, 2024) | Zero Shot Learning | Visual Question Answering |
| | ZPVQA (Hu et al., 2025a) | Zero Shot Learning | Visual Question Answering |
| | Prophet (Yu et al., 2025b) | Efficient Fine Tuning | Visual Question Answering |
| | PromptCAP (Hu et al., 2023) | Efficient Fine Tuning, Zero Shot Learning | Visual Question Answering |
| | Img2LLM (Guo et al., 2023) | Zero Shot Learning | Visual Question Answering |
| | RQP (Lan et al., 2023) | Zero Shot Learning | Visual Question Answering |
| | ViDIL (Wang et al., 2022) | Few Shot Learning | Video Lanugage Tasks |
| | cross-modal adaptation (Lin et al., 2023b) | Few Shot Learning | Image Classification |
| | PNP-VQA (Tiong et al., 2022) | Zero Shot Learning | Visual Question Answering |
| | F-VQA (Wu et al., 2023c) | Zero Shot Learning | Visual Question Answering |
| | R2A (Pan et al., 2023) | Zero Shot Learning | Video Question Asnwering |
| | LADS (Dunlap et al., 2023) | domain adaptation | Image Classification |
| | Cross-Modal Adaptation (Lin et al., 2023b) | Few Shot Learning | Image Classification |
| | Z-LaVI (Yang et al., 2022c) | Zero Shot Learning | language understanding |
| | protext (Khattak et al., 2024) | Efficient Fine Tuning | Image Classification |

Table 4: Transfer learning methods based Knowledge

mapper to extract vision-language representation token. The two tokens, along with task instruction and text features, are then sent to a multimodal fusion mapper for alignment across different modalities. Similarly, Chen et al. (2022b) use a V-L mapper that maps the visual features into the language feature space, which allosw the vision and language module to be progressively and independently pre-trained. Mañas et al. (2022) propose a method to adopt pre-trained unimodal models for VL tasks without intensive training or finetuning. It directly adopts a frozen pre-trained vision encoder and frozen LLM, connecting them with a trainable mapping network, which projects the visual feature into the dimension of text features. The mapping network is then updated with an image captioning objective. Liu et al. (2021) propose a cross-modal interaction method for composed image retrieval. While the images are encoded through ResNet, the reference image is sent to the VLM, OSCAR (Li et al., 2020), along with the tokenized text. After alignment within the VLM, the image feature token is compared with the candidate target image feature tokens to select the top match. The cross-modal methods are summarized in Table3 cross-modal interaction section.

Some methods, similar to adapter-based and prompt-based methods, propose additional novel modules to the VLM, but these novel modules are different from adapter, prompts or any other seen methods. For example, McDonnell et al. (2024) adopt random projection for continual learning tasks. The random projection layer, along with non-linear activation function, is inserted between the pre-trained model's feature extraction and output head in order to capture interactions between features with expanded dimensionality, providing enhanced linear separability for class-prototype-based CL. Yu et al. (2023) introduce a TaskRes for expanding the classifier for efficient transfer learning. Different from using prompts or adapters that update the classifier through training, TaskRes freezes the whole model, including the classifiers. It adds tunable parameters which are not dependent on based classifier, directly to the classifier by weighted sum to achieve better old knowledge inheritance while flexible task-specific knowledge exploration. Ni et al. (2024) propose LingoCL which relies on the significance of the semantic information within the label names while others encode label as one-hot label. LingoCL uses a pre-trained language model to generate semantic target based on label name, and uses the output features as the weights for the frozen classifier, guiding the learning of encoders.

Other than creating novel modules, Zhang et al. (2023c) introduce a few-shot learning pipeline, CaFo, by cascade combination of the prior-knowledge from four existing pre-trained models. Given the input image-text pair, CaFo firstly utilizes GPT-3 to generate diverse textual descriptions of the input image, and uses DALL-E to generate synthesized few-shot training images which are used to build cache of prior-knowledge with the visual output from both CLIP and DINO. During inference, the target image is sent to both CLIP and DINO to generate visual features, which are used to retrieve knowledge from cache for prediction.

## 4.4 Knowledge-Based Methods

Different from all the methods above, which rely on improving the model to better understand the input semantics for better performance, the knowledge-based methods leverages pre-existing knowledge from large-scale pre-trained models or external sources of knowledge to improve performance on new, unseen tasks or domains that combine visual and textual data. The diagram shown the general architecture can be found in

Figure 6, in which the inputs are sent to a pre-trained LLM to generate supplement knowledge to enhance the semantic information.

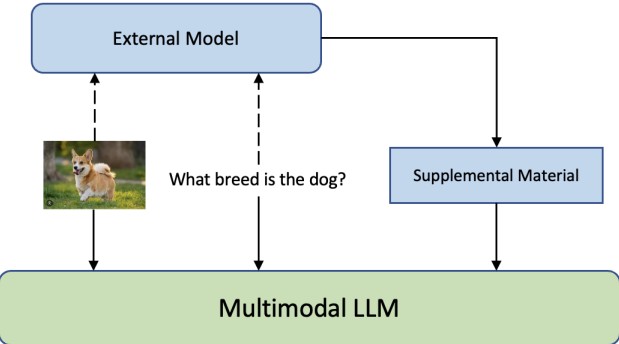

Figure 6: Taxonomy of knowledge-based Methods. For knowledge-based methods, the input image, text, or both, is sent to an external LLM to retrieve the supplemental knowledge. The supplemental knowledge are then set to the model to help the comprehensive understanding of task.

Knowledge-based methods are widely used in visual question answering (VQA) tasks. Cao & Jiang (2024) propose KGenVQA to enable the zero-shot learning for VQA. KGenVQA converts the image into a text description via a LLM, and then uses the LLM-generated description and question to form the initial knowledge. The selected knowledge, along with the question and description, are input to the LLM for answer generation. Hu et al. (2023) propose a method that generates captions based on the image and provide it as additional knowledge to align the different modalities to improve the model's performance on VQA tasks. Hu et al. (2025a), similarly, augment the text input with the LLM's generation. It generates caption prompts from images, along with samples of synthesized question-answer pairs to serve as example prompts in context, thus enhancing the model's understanding of the associations between the images and the questions. Yu et al. (2025b) introduce a framework, Prophet, designed to prompt LLM with answer heuristics for knowledge-based VQA. Prophet first trains a vanilla VQA model on a specific knowledge-based VQA dataset without external knowledge. Then, it extracts two types of complementary answer heuristics from the VQA model: answer candidates and answer-aware examples. The two types of answer heuristics are jointly encoded into a formatted prompt to facilitate the LLM's understanding of both the image and question, thus generating a more accurate answer. Khan et al. (2024) propose a question decomposition strategy to provide external knowledge and offer second-guessing for a LLM to answer the question. Given the question and the image, they first decompose the question into several sub-questions and provide a sub-answer for each of the questions. The generated sub-questions and sub-answers are served as the context, along with original question, of the input to the LLM. Guo et al. (2023) offer an additional text input to the text encoder by generating captions using their proposed Img2LLM module. The Img2LLM generates an image caption based on a text-guided image feature and exemplar question-answer pair. Similarly, Lan et al. (2023) rely on image-guided caption generation for solving zero-shot VQA tasks. The method contains two stages. On the first stage, the image caption is generated, and the caption then is combined with the question to generate image-guided questions and the answer accordingly. On the second stage, the generated answer with its corresponding confidence score is used as the answer heuristic to guide the final output. Tiong et al. (2022) introduce plug-and-play VQA (PNP-VQA). It contains an image-question matching module to generate the attention map on the image guided by question, and adopts the attention-enhanced image to generate a set of captions. The most related captions are combined with the question into the question-answering module for answer generation. Liu et al. (2024a) propose ZVQAF. ZVQAF finetunes a caption model for external knowledge. During the training stage, it uses a frozen image-text matching module to select relevant image regions, and generates a caption based on the regions. While the caption and the original question are sent to the frozen LLM to generate the answer, the captioning model is updated via reinforcement learning strategy. Wu et al. (2023c) utilize multiple external knowledge to generate answer. The proposed method identifies the "relation" and "entity" part from the text and embeds them into separate spaces, and aggregates the feature in different spaces with a weighted score for answer prediction. Other than focusing on image-based VQA tasks, some knowledge-based methods help improve the model's performance on the video question

understanding, instead of the image. Wang et al. (2022) propose a few-shot learning method for video question answering. Given the video input, it extracts the visual semantics from the token level, the frame level, and the video level by converting them into the textual representation, and the textual description is fed into the language model with task-specific instruction to generate answer. Pan et al. (2023), on the other hand, adopt the CLIP model to retrieve texts from the text corpus that are relevant to the video frames, then input both the retrieved text and the question, along with video visual features, into the LLM.

Moreover, the knowledge-based methods are not constrained to the visual question answering tasks. Dunlap et al. (2023) propose a knowledge-based method for unseen image domain adaptation, namely Latent Augmentation using Domain Descriptions (LADS). Given an image from the source domain and a text description from both the source and the target domain, the method introduces an augmentation network to transform image embeddings from the source to the target domain using a domain alignment loss and a class consistency loss between the texts from two domains to ensure that the shifted image feature is in a new domain while it remains in the same class. Lin et al. (2023b) develop a knowledge-based method for image classification. While image classification is a uni-modal task, the method includes additional modalities as inputs, e.g., text and audio. These additional modalities are seen as the additional few-shot examples. By mapping inputs from different modalities into the same representation space, the knowledge from the additional modalities can serve as the supplementary information for the image modality and help the classification task. Cai et al. (2025) propose a training-time negation data generation strategy for CLIP that dynamically constructs negated captions without relying on large LLM-generated datasets. By exposing models to diverse, non-stationary negation examples during contrastive training, it improves negation awareness several tasks. Yang et al. (2022c) propose Z-LAVI that adopts external visual knowledge for uni-modal text tasks. Given a text input, either the corpus or the labels, Z-LAVI retrieves an image by recalling existing images from a search engine and generates synthesized images using visual generative models. The result of the LLM and the VLM are assembled to predict the final answer. Lastly, (Khattak et al., 2024) propose a knowledge-based prompt learning method, *protext*, for recognition task. Instead of directly freezing the model and train the text prompt, *protext* first collects the detailed descriptions for training classes via LLM, and then use contrastive learning to align the image caption with corresponding descriptions to enrich the text encoder with diverse contextual knowledge. All the methods above are summarized into Table 4.

# 5 Application Areas of Transfer Learning for Vision & Language Models

While efficient transfer learning methods significantly reduce the computational and data requirements for adapting large VLMs, their true impact is best demonstrated through real-world applications. In this section, we explore how these transfer learning methods enable practical and scalable implementations of VLMs, showcasing their role in solving real-world vision-language challenges, in medical, robotic, and human understanding domain, while maintaining efficiency and adaptability.

## 5.1 Medical & Health Care

Pre-trained VLMs, such as Flamingo, LLaVA, CLIP, etc., have been adopted in medical field by finetuned on medical data(Chambon et al., 2022; Chen et al., 2022c; Moor et al., 2023; Li et al., 2024a). However, since medical data is often scarce, compared with the general domain data, more advanced transfer learning plays a significant role in building effective medical VLMs.

A group of methods adopt prompt-based techniques for classification of medical image (Chang et al., 2024; Wang et al., 2024; Ye et al., 2024; Denner et al., 2024). Those methods explore the zero-shot and the few-shot learning using VLMs on radiology and X-ray images. Lu et al. (2023), on the other hand, introduce a multiple-instance learning technique to solve zero-shot image classification on gigapixel images, enhancing the capacity of CLIP capacity on large-scale medical image. Shakeri et al. (2024) introduce a few-shot medical image classification benchmark that can be used for validation and model improvement.

VLMs have also been used on medical image segmentation tasks. Poudel et al. (2024) finetune a general-purpose VLM with segmentation capability and evaluate its performance on 11 diverse medical datasets. Wang et al. (2023c) adopt a prompt-based technique for medical image segmentation by introducing Fourier

visual prompts to enhance the model's performance on X-ray images. Jiang et al. (2024) explore zero-shot lesion segmentation on 3D medical images with cross-modal interaction using VLMs.

Another group of methods explores text generation in the medical domain (Wu et al., 2023d; Li et al., 2024b). Wu et al. (2023d) adopt adapters and medical knowledge enhancement loss to improve medical report generation based on medical images. Li et al. (2024b) improve the quality of multimodal medical dialogue by extracting the relevant region of an input image in a zero-shot manner. Yu et al. (2025a); Gu et al. (2024), on the other hand, propose propose the prompt-based method to adopt VLMs for medical-VQA tasks for organ-level disease recognition and radiology analysis.

## 5.2 Robotic System

VLMs have been used to enable robots to better understand and interact with their environments through multimodal reasoning. Kuzmenko & Shvai (2024) examine variant VLMs on zero-shot navigation tasks and study their efficiency for planning and navigation. Chen et al. (2024a) propose a method to adopt commonsense knowledge of VLMs for a legged robot to deal with difficult and ambiguous real-world navigation situations. Narasimhan et al. (2024) introduce a lifelong learning method for VLMs to enhance navigation performance of service robots in human-centered environments. Unlu et al. (2024) utilize GLIP and instruct-BLIP for zero-shot object goal creation, by letting GLIP to target the object and then use instructBLIP to confirm it. Venkatesh & Min (2024), on the other hand, introduce a VLM for multi-robot pattern formation by using the knowledge from the pre-trained model to translate natural language instructions into the actionable robot configurations.

In addition to robotic navigation, Liu et al. (2024b) apply mark-based visual prompts on VLMs to achieve open-world robotic manipulation via free-form natural language information. Khorramshahi et al. (2021), on the other hand, adopt CLIP's zero shot capability to build a vehicle re-identification retrieval system to identify the make, the model, the color, and the production year of a given vehicle.

## 5.3 Human Understanding

Human understanding tasks include a variety of machine learning challenges focused on interpreting and analyzing human-related attributes, behaviors, and interactions from visual and multimodal data. With the adoption of VLMs and the corresponding transfer learning strategies, human understanding tasks can be tackled in a more efficient and effective way.

Wang et al. (2023a) finetune the CLIP model with a video contrastive loss, and use a temporal transformer to refine the support videos with the textual information guide for few shot learning in action recognition. Shi et al. (2024) also adopt the CLIP model, and collect a corpus of an action-related text database from an external source to guide the few-shot recognition with commonsense knowledge. Xing et al. (2024b) enhance the CLIP vision encoder with a global temporal adapter and local multimodal adapter, and freeze the rest of the model during finetuning for efficient few-shot action recognition adaptation. Lu et al. (2024), on the other hand, focus on the skeleton action recognition. They use the VLM to help the pre-training of the skeleton encoder, and train a cosine classifier as the support set. Wang et al. (2023b) adopt a text caption model, e.g., BLIP, to align the support and query action recognition videos in a cross-modal approach. Meanwhile, Lin et al. (2023a) and Phan et al. (2024) propose methods for zero-shot action recognition based on VLMs for improving the model's generalizablity on unseen domain.

The efficient VLM transfer learning has also been applied for the facial expression recognition tasks. Foteinopoulou & Patras (2024) and Zhao et al. (2024) introduce the variants of CLIP by training CLIP with the sample-level text descriptions and videos, and inference using the class name, (e.g., happiness and sadness), for the zero-shot facial expression recognition task. Zhao & Patras (2024) also utilize CLIP by adding the learnable context tokens to the text encoder and attach temporal module after vision encoder. Zhang et al. (2024d) propose Set-of-Vision prompting which attaches spatial information such as bounding boxes and facial landmarks to VLM's visual input to enable zero-shot emotion recognition. Chen et al. (2025b), on the other hand, design a mixture-of-expert (MoE) adapter architecture on top of the VLM. It

| Category | Dataset | Training | Testing | Domain |
|---|---|---|---|---|
| Image Classification | MNIST(Lecun et al., 1998) [link] | 60,000 | 10,000 | handwriting number |
| | ImageNet-1K (Deng et al., 2009) [link] | 1,281,167 | 50,000 | general |
| | CIFAR-10 (Krizhevsky et al., 2009) [link] | 50,000 | 10,000 | animal, transportation |
| | CIFAR-100 (Krizhevsky et al., 2009) [link] | 50,000 | 10,000 | real-world objects |
| | SUN397 (Xiao et al., 2014) [link] | 19,850 | 19,850 | general |
| | Stanford-Cars (Krause et al., 2013) [link] | 8,144 | 8,041 | vehicles |
| | Oxford 102 Flowers (Nilsback & Zisserman, 2008)[link] | 2,040 | 6,149 | flowers |
| | Food-101 (Bossard et al., 2014) [link] | 75,750 | 25,250 | food |
| | Stanford Dogs Khosla et al. (2011) [link] | 12,000 | 8,580 | dogs |
| | Fashion-MNIST (Xiao et al., 2017) [link] | 60,000 | 10,000 | fashion |
| | SVHN (Xiao et al., 2023) [link] | 73,275 | 26,032 | house numbers |
| | Caltech-101 (Li et al., 2022a) [link] | 3,060 | 6,085 | general |
| | FGVC-Aircraft (Maji et al., 2013) [link] | 6,667 | 3,333 | aircraft |
| | Birdsnap (Berg et al., 2014) [link] | 42,283 | 2,149 | birds |
| | Places (Zhou et al., 2018a) [link] | 1,800,000 | 328,500 | city scene |
| | CelebA (Liu et al., 2015) [link] | 162,770 | 19,962 | face attributes |
| Image Captioning | Flickr8K (Plummer et al., 2015) [link] | 7,000 | 1,000 | general |
| | Flickr30k (Plummer et al., 2015) [link] | 31,728 | - | general |
| | COCO Caption (Chen et al., 2015) [link] | 82,783 | 5,000 | general |
| | Visual Genome (Krishna et al., 2016) [link] | 108,000 | - | general |
| | AI2D (Kembhavi et al., 2016) [link] | 4,000 | 1,000 | scientific diagram |
| | CUB-200 (Reed et al., 2016) [link] | 5,994 | 5,794 | birds |
| | Fashion Cap. (Yang et al., 2022b) [link] | 993,000 | - | fashion |
| | CC3M (Sharma et al., 2018) [link] | 3,300,000 | - | general |
| | CC12M (Changpinyo et al., 2021) [link] | 12,400,000 | - | general |
| | TextCaps. (Sidorov et al., 2020) [link] | 21,953 | 3,289 | general |
| Image-Text Retrieval | Flickr30K (Plummer et al., 2015) [link] | 31,728 | - | general |
| | Visual Genome (Krishna et al., 2016) [link] | 108,000 | - | general |
| | MSCOCO (Lin et al., 2014) | 82,783 | 5,000 | general |
| | WIT (Srinivasan et al., 2021) [link] | 37,600,000 | - | Wikipedia |
| | Open Image (Kuznetsova et al., 2020) [link] | 8,850,000 | 125,000 | general |
| Visual Question Answering | VQAv2 (Goyal et al., 2017) [link] | 83,000 | 81,000 | general |
| | CLiMB (Srinivasan et al., 2022) [link] | 83,000 | 81000 | general |
| | GQA (Hudson & Manning, 2019) [link] | 113,000 | - | scene graph |
| | OKVQA (Marino et al., 2019) [link] | 14,000 | - | external knowledge |
| | COCOQA (Ren et al., 2015) [link] | 123,000 | 78,000 | general |
| | TextQA (Singh et al., 2019) [link] | 18,408 | - | text material |
| | TDIUC (Kafle & Kanan, 2017) [link] | 167,000 | - | general |
| | KVQA (Shah et al., 2019) [link] | 183,000 | - | knowledge graph |
| Visual Entailment | SNLI-VE (Xie et al., 2019) [link] | 529,000 | 17,000 | general |
| | NLVR2 (Suhr et al., 2019) [link] | 86,000 | 10,200 | general |
| | Flickr30k Entities (Plummer et al., 2016) [link] | 244,000 | - | general |
| | Hatefull Memes Dataset (Kiela et al., 2021) [link] | 10,000 | - | hateful content detection |
| Semantic Segmentation | Cityscapes (Cordts et al., 2016) [link] | 3,475 | 1,525 | urban scene |
| | PASCAL VOC (Everingham et al., 2010) [link] | 11,540 | - | general |
| | ADE20K (Zhou et al., 2018b) [link] | 20,210 | 3,000 | general |
| | COCO-stuff (Caesar et al., 2018) [link] | 164,000 | - | general |
| | Mapillary Vistas (Neuhold et al., 2017) [link] | 25,000 | - | street scene |
| | CamVid (Brostow et al., 2008) [link] | 700 | - | road scene |
| | BDD100K (Yu et al., 2020) [link] | 100,000 | - | driving |

Table 5: Summary of commonly used datasets

creates one adapter for each modality of the input to captures and fuses emotional movements from different data.

As the efficient VLM transfer learning methods continue to demonstrate strong performance across real-world application domains, it becomes increasingly clear that progress in this area depends not only on architectural innovation but also on the quality of the data used to train and evaluate these models. High-quality, diverse, and well-curated datasets are essential for capturing the complexity of vision-language interactions and for ensuring that improvements translate beyond controlled settings. In the following section, we introduce the key datasets and benchmarks for the VLM researches and provide the foundation for measuring progress in vision-language learning.

# 6 Vision Language Datasets & Benchmarks

Vision-language datasets play a pivotal role as the foundational backbone for training and evaluating VLMs. These datasets are designed to cover a wide range of challenges, from basic detection and classification tasks to visual–linguistic reasoning tasks that mimic complex human cognitive abilities. Beyond enabling model

training, dataset diversity and coverage critically influence model generalization, as insufficient or biased data distributions can lead to phenomena such as mode collapse Hu et al. (2025b), where models overfit to dominant patterns while failing to represent rare or diverse multimodal concepts. Vision-language datasets therefore not only enable training VLMs but also shape how effectively they understand and interact with multimodal inputs. In this section, we categorize vision-language datasets into six major task types, with a summary of all datasets presented in Table 5.

**Image Classification**  task aims to assign predefined labels to images based on their visual content. Data in image classification dataset are organized as image-label pairs:

$$\mathcal{D}_i = \{I_i, Y_i\}, \tag{7}$$

Where $\mathcal{D}_i$ denotes the arbitrary sample of an image classification dataset, and $I_i$, $Y_i$ represent the image and label in the sample. Image classification datasets are different in terms of size, number of classes, domains, etc. Image classification tasks are mainly evaluated via accuracy metric.

**Image Captioning**  task generates descriptive textual sentences for images by understanding their visual content and context. Data pairs are presented in the format of:

$$\mathcal{D}_i = \{I_i, C_i\}, \tag{8}$$

where $C_i$ is the image caption. Different from image classification task that can be simply measured by accuracy, image captioning are evaluated through more complex benchmarks such as BLEU (Papineni et al., 2002), CIDEr (Vedantam et al., 2015), and SPICE (Anderson et al., 2016), etc.

**Image-Text Retrieval**  datasets are deployed to align visual and textual modalities to enable the retrieval of the relevant image given the textual query (image retrieval) or the relevant text given an image query (text retrieval). For image-text retrieval datasets, data is organized similar to image captioning:

$$\mathcal{D}_i = \{I_i, C_i\}. \tag{9}$$

However, a given model needs to learn a reliable matching pattern between the two modalities. The evaluation metric for image-text retrieval is primarily *Recall@K*, which measures the proportion of relevant items in the top-K results of the ranking given the query.

**Visual Question Answering**  task combines computer vision and natural language processing to answer questions about the content of an image. It requires models to understand both the image and the question, and to reason about objects, relationships, and contextual details. The data for VQA is in a triplet form:

$$\mathcal{D}_i = \{I_i, Q_i, A_i\}, \tag{10}$$

where $Q_i$ represents the question and $A_i$ represents its answer. Visual question answering tasks can also be evaluated using accuracy, BLEU and F1 score.

**Visual Entailment**  task is the multimodal challenge that involves determining whether a given text, denoted as hypothesis, logically entails, contradicts, or is neutral to an image. In this task, a model is presented with an image and a text statement, and it must tell the relationship in one of the three categories. Similar to the visual question answering data sample, the visual entailment data is in a triplet form:

$$\mathcal{D}_i = \{I_i, H_i, Y_i\}, \tag{11}$$

where $H_i$ means the hypothesis. The model takes the image and the hypothesis as inputs and predicts the label within entailment, neutral and contradiction. The most common metric for visual entailment task is accuracy.

**Semantic Segmentation**   is a computer vision task that involves classifying each pixel in an image into a predefined set of categories, and effectively partitioning the image into segments that correspond to different objects or regions. Semantic segmentation datasets are organized in the form of:

$$\mathcal{D}_i = \{I_i, M_i\}, \tag{12}$$

where $M_i$ denotes the semantic mask that associates each pixel value with a class label. The common metric for semantic segmentation task are pixel accuracy and intersection over union (IoU).

## 7   Conclusion

In this paper, we explored the landscape of efficient transfer learning methods for vision-language tasks, highlighting the state-of-the-art approaches that leverage adapters, prompts, innovative model and external knowledge. These methods have demonstrated significant advancements in enabling pre-trained models to adapt effectively to a wide range of vision-language applications, from image captioning to visual question answering, with improved the VLM efficiency in terms of computational resources and data usage. Transfer learning methods are essential for advancing VLMs, enabling bridging the gap between visual and textual understanding in a cost-effective manner. As the field evolves, continued innovation in transfer learning techniques will be instrumental in unlocking new possibilities for VLMs across diverse domains.

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
