# OpenReview forum: "Efficient Adaptation of Large Vision-Language Models: Transfer Learning Methods and Applications"
_TMLR — Rejected by TMLR_

### Review · Reviewer_9DPc · 2026-02-14

**Summary Of Contributions:**

## Summary Of Contributions

This paper presents a literature survey on transfer learning techniques for large vision-language models (VLMs), organized into four main components: (1) an overview of state-of-the-art VLMs, (2) a taxonomy of transfer learning approaches for VLMs, (3) applications of these methods in real-world scenarios, and (4) a summary of commonly used datasets and benchmarks for vision-language tasks.

### Strengths
- Addresses a timely and important topic given the rapid development of VLMs
- Attempts to provide a comprehensive overview of an active research area

### Weaknesses
- Lacks systematic methodology for paper selection and inclusion criteria
- Lacks reproducibility—no codebase provided for method comparison and evaluation
- Contains factual inaccuracies and incomplete citations (see detailed comments below)
- Insufficient engagement with critical topics (e.g., catastrophic forgetting)
- Multiple spelling and citation errors that impact readability and credibility

**Audience:**

No

**Audience Explanation:**

While the topic itself is relevant to the TMLR community, the current execution falls short of publication standards:

- The survey in its present form does not meet the threshold for a venue like TMLR, which prioritizes rigorous, reproducible research with clear contributions. A high-quality survey should either provide novel organizational insights, systematic comparisons, or reproducible benchmarks.
- The factual errors, incomplete citations, and methodological ambiguities suggest the paper requires substantial revision before it would be suitable for publication.
- Readers would benefit more from a survey that includes: (1) transparent selection methodology, (2) comparative empirical analysis or critical synthesis, (3) accurate and complete citations, and (4) reproducible resources.
- In its current form, the paper is more likely to mislead than to inform the audience due to the factual inaccuracies noted above.

**Claims And Evidence:**

No

**Claims Explanation:**

The paper has several critical issues that undermine its credibility as a comprehensive survey:

**Methodological Concerns:**
- The paper lacks a clearly stated, reproducible methodology for selecting which papers to include. There is no explicit timeline, inclusion/exclusion criteria, or justification for the selection of surveyed works.
- Without these details, it is impossible to assess whether the survey represents the landscape comprehensively or contains selection bias.
- The paper presents summaries of existing methods but provides no comparative analysis, critical evaluation, or discussion of the individual strengths and weaknesses of different approaches.

**Factual and Technical Errors:**
- The claim that domain adaptation "only has unlabeled data" is factually incorrect and ignores the well-established field of supervised domain adaptation.
- The paper provides minimal discussion of catastrophic forgetting, which is a fundamental concern in transfer learning and fine-tuning. Critically, the seminal work "Overcoming Catastrophic Forgetting" (Kirkpatrick et al., 2016) is not cited, suggesting incomplete coverage of foundational concepts.
- The paper contains spelling errors (e.g., "test to image generation" instead of "text-to-image generation") that detract from clarity and professionalism.

**Citation and Attribution Issues:**
- The citation "Learning multiple layers of features from tiny images" contains the placeholder "[Insert the date you accessed this paper]" rather than an actual publication date, indicating incomplete citation work.
- "Learning to Prompt with Text Only Supervision for Vision-Language Models" is cited as an arXiv paper when it has been accepted at AAAI, representing a failure to cite the work in its correct published venue.
- These errors suggest inadequate review and fact-checking before submission.

**Lack of Reproducibility:**
- A comprehensive survey on transfer learning methods should include a reproducible codebase and empirical comparisons of the surveyed methods. The absence of such materials limits the utility and credibility of the survey.
- Without comparative experiments, the paper cannot substantiate claims about the relative merits of different approaches or guide practitioners in selecting appropriate methods.

**Requested Changes:**

1. **Define and justify selection methodology**: Clearly state the search strategy, inclusion/exclusion criteria, and time period covered. Justify any papers excluded and explain how the selection avoids bias.

2. **Correct factual errors**: Revise claims about domain adaptation, catastrophic forgetting, and other technical concepts. Include seminal works like Kirkpatrick et al. (2016) and other foundational papers.

3. **Add comparative analysis**: Either through systematic tables comparing method characteristics, or through empirical benchmarking on standard datasets, provide substantive comparison of the surveyed approaches.

4. **Complete and verify citations**: Ensure all citations are accurate, complete, and use the correct publication venues. This includes replacing all placeholder text.

5. **Provide reproducible resources**: Develop and share a codebase that implements or benchmarks the surveyed methods, enabling readers to verify claims and build upon the work.

6. **Conduct thorough proofreading**: Correct spelling and grammatical errors throughout.

7. **Strengthen coverage of critical topics**: Expand discussion of catastrophic forgetting, domain shift, and other central challenges in transfer learning for VLMs.

---

> ### Author Response · Authors · 2026-02-16
> **Clarifying questions**
>
> Dear reviewer,
>
> We are grateful for your time and review. We appreciate that you have found work a timely survey in the field. Recognizing weaknesses, we are glad that you have provided a pathway to improve our work and we are hopeful that we can achieve the quality level that you will find acceptable. We have a few questions about your review:
>
>
> 1. We do agree that there can be works that we have missed to include. We are grateful if the reviewer could include the list of works that you deem important but we have missed in our work.
>
> 2.  "A comprehensive survey on transfer learning methods should include a reproducible codebase and empirical comparisons of the surveyed methods. The absence of such materials limits the utility and credibility of the survey."
>
> We agree that reproducibility and empirical benchmarking are highly valuable contributions to the community. However, we would respectfully note that the primary goal of a survey paper is to provide a structured synthesis of existing literature, identify trends, clarify conceptual distinctions, and highlight open challenges, rather than to re-implement and benchmark the surveyed methods. Reproducing a comprehensive set of all methods that we have surveyed would require substantial computational resources and experimental standardization across diverse model scales, datasets, and evaluation protocols, which typically falls outside the scope of a survey paper. That said, we understand the reviewer’s concern regarding practical utility. To strengthen the manuscript, we would be happy to:
>
> - Add a detailed comparative summary table consolidating reported experimental results from the original papers.
>
> - Clearly summarize evaluation settings and datasets to improve reproducibility transparency.
>
> - Include references to publicly available implementations for the surveyed methods.
>
> We would appreciate clarification on whether the reviewer's concerns would be addressed by doing the above? If not, could you please what do you expect to address the above concern?
>
>
> 3. "Without comparative experiments, the paper cannot substantiate claims about the relative merits of different approaches or guide practitioners in selecting appropriate methods."
>
> We respectfully note that, as a survey paper, our intention is not to claim new empirical superiority of specific methods, but rather to synthesize and critically analyze findings reported in the existing literature. Direct comparative experiments across all the surveyed methods is extremely challenging due to differences in model scale, pretraining corpora, downstream tasks, evaluation metrics, and compute budgets reported across studies. However, we agree that clearer practitioner guidance would strengthen the paper. To address this concern, we propose to:
>
> - Add a decision-oriented summary section outlining when each class of methods is most appropriate.
>
> - Provide structured comparison tables summarizing reported results for a similar subset of works that we have covered which makes comparing them feasible. We will mention explicitly that comparisons are drawn from reported results rather than newly conducted experiments.
>
> We would appreciate clarification on whether the reviewer's expectations would be satisfied by the above?
>
> 4. Recognizing your time limit, we are grateful if you can mention any concrete changes that you have in mind for making our work acceptable.

---

> > ### Comment · Reviewer_9DPc · 2026-02-16
> > **clarification**
> >
> > 1. Table 1 lists 18 methods spanning 2021–2025, yet provides **no explicit selection criteria** for how these works were identified or chosen.
> >
> > Given the significant expansion of CLIP-based adaptation research during this period, 18 works represents a substantial undersampling of the field. The omission of major contributions from flagship conferences raises serious concerns about the comprehensiveness of the survey:
> >
> > ### Missing Key Works (Examples)
> > - Goyal, Sachin, et al. "Finetune like you pretrain: Improved finetuning of zero-shot vision models." *Proceedings of the IEEE/CVF Conference on Computer Vision and Pattern Recognition*. 2023.
> > - Wei, Yixuan, et al. "Improving clip fine-tuning performance." *Proceedings of the IEEE/CVF International Conference on Computer Vision*. 2023.
> > - Zheng, Zangwei, et al. "Preventing zero-shot transfer degradation in continual learning of vision-language models." *Proceedings of the IEEE/CVF International Conference on Computer Vision*. 2023.
> > - Zhang, Wenxuan, et al. "Overcoming generic knowledge loss with selective parameter update." *Proceedings of the IEEE/CVF Conference on Computer Vision and Pattern Recognition*. 2024.
> > - Lai, Zhengfeng, et al. "Clipath: Fine-tune clip with visual feature fusion for pathology image analysis towards minimizing data collection efforts." *Proceedings of the IEEE/CVF International Conference on Computer Vision*. 2023.
> > - Lu, Ziqian, et al. "Improving zero-shot generalization for clip with variational adapter." *European Conference on Computer Vision*. Cham: Springer Nature Switzerland, 2024.
> >
> > **Critical Issues:**
> > - Omission of CVPR, ICCV, and ECCV papers—the premier venues in this area—contradicts claims of comprehensiveness
> > - No justification for excluding these prominent works
> > - For a survey paper without code or experimental contributions, authors bear the responsibility to conduct exhaustive literature searches across all relevant venues
> > - The burden of literature discovery should not **fall on reviewers**
> >
> > **Required Action:** The authors must provide explicit inclusion/exclusion criteria and substantially expand the survey to capture the full scope of published work.
> >
> > **Table 1** claims to cover "different domains" across 18 adapter-based methods (2021–2025), yet the coverage is not clearly stratified by domain or application context.
> >
> > **Table 4** is titled "Transfer learning methods based Knowledge," which is not a grammatically correct or informative title. Moreover, Table 4 does not provide the multi-domain coverage(eg: continual learning) present in Table 1, suggesting inconsistent scope and organization.
> >
> > **Recommendation:**
> > - Clarify the domain coverage for each table
> > - Improve table titles to be grammatically correct and descriptive
> > - Ensure consistent scope across related tables
> >
> > 2.
> > Per TMLR guidelines for survey papers:
> >
> > > "Authors should make sure to emphasize the contributions made by the survey. Ideally, we want survey papers that draw **new, previously unreported** connections between several pieces of work in an area, and/or that clearly highlight trends in the area and/or suggest currently **open problems.**"
> >
> > A simple enumeration of some methods does not meet this standard. The authors must provide substantive, novel contributions beyond summarization.
> >
> > ### Reference Examples
> > - [TMLR Accepted Survey Example](https://openreview.net/pdf?id=bsCCJHbO8A): Maintains a curated repository of papers with linked codebases
> > - [ICCV Workshop example](https://github.com/AlbinSou/ocl_survey): Authors provide comprehensive experimental setup across surveyed methods.
> >
> > ### Required Contributions in addition to what authors provided.
> > Given the paper's focus on fine-tuning Vision-Language Models (VLMs) with benchmark and dataset resources, the authors should consider providing:
> > - **A maintained GitHub repository** indexing papers with associated codebases and reproducibility information
> > - **Clear identification of research gaps** and promising future directions

---

> > > ### Comment · Reviewer_9DPc · 2026-02-16
> > > **clarification**
> > >
> > > 3. I appreciate the authors’ clarification that the manuscript is intended as a survey rather than an empirical study. However, precisely because the paper does not introduce new experimental results, this substantially heightens expectations in other dimensions.
> > >
> > > In particular, a survey without experimental contribution must demonstrate:
> > >
> > > - **Spelling accuracy and grammatical correctness**
> > > - **Precise, factually accurate definitions** (e.g., domain adaptation)
> > > - **Correct citations**
> > >
> > > At present, these standards are not consistently met.
> > >
> > > Moreover, while I understand the difficulty of reproducing large-scale pretraining results, this paper focuses on fine-tuning methods for VLMs, not on pretraining from scratch. This significantly lowers the barrier to constructing a unified benchmark framework.
> > >
> > > Even if full-scale reproduction of all surveyed methods is infeasible, the authors could:
> > > - Select a representative pretrained VLM backbone,
> > > - Implement or adapt a subset of widely used fine-tuning strategies,
> > > - Evaluate them under a standardized protocol (datasets, metrics, compute budget),
> > > -Provide controlled comparisons within this unified setting.
> > >
> > > Such a standardized comparison would meaningfully strengthen practitioner guidance and elevate the paper beyond a descriptive summary of some prior work.
> > >
> > > Without either: New comparative evidence in a controlled setting, or Exceptionally rigorous and precise synthesis,
> > > the manuscript risks lacking the level of clarity expected of a survey in this area.

---

> > > > ### Author Response · Authors · 2026-02-16
> > > >
> > > > We are thankful for your continual engagement and appreciate the detailed comments. We will definitely consider them well when updating the manuscript.

---

### Review · Reviewer_yPha · 2026-03-20

**Summary Of Contributions:**

This paper presents a literature review of transfer learning (TL) approaches for Vision-Language Models (VLMs). The paper starts with a brief overview of VLM architectures and optimization objectives, followed by a discussion of various TL methods. Finally, the authors detail relevant application domains, dataset benchmarks, and multi-modal tasks.

Weaknesses

Currently, the paper reads more like an unstructured list of TL approaches rather than a cohesive review, making it difficult to find a central conclusion or take away message. A strong literature review should go beyond merely summarizing individual papers. It should synthesize the literature to derive new insights, identify open problems, propose future research directions, or provide actionable guidance (e.g., a practical framework detailing which TL method to use in specific situations). The paper currently lacks this any level of analysis.

Several statements are presented as factual assumptions without appropriate citations or justifications. The authors must ensure all claims are either properly cited or sufficiently explained. Some examples include:

“... they generally lack fine-grained cross-modal reasoning since the encoders operate independently.” This is stated as a fact but lacks a supporting reference. Please cite a source that demonstrates this limitation.

“... the massive scale introduces significant computational and deployment costs” It is unclear how transfer learning reduces deployment costs, given that the final model's total parameter size remains the same as far as I know TL primarily reduces future training costs not the storage costs.

“...Since only a small number of parameters are updated during finetuning, adapter-based methods significantly reduce the memory requirement for model storage .” This claim requires more context. While adapter methods save space when storing multiple task-specific updates, the full base model must still be stored on disk.

**Additional Comments:**

- Please use \citep{} consitently everywhere.
- The definition of Mask Language Modelling, eq. (4), is wrong.  This is the definition of next token prediction (since there is a casaul depenencies on t-1,...1) and not on the entire context as with MLM.

**Audience:**

No

**Audience Explanation:**

The manuscript currently reads more like an introductory textbook for a deep learning course than a research-level survey.
A literature review must go beyond summarizing current literature; it should synthesize the combined literature to extract novel insights, identify trends, or draw overarching conclusions. These critical elements are missing from the current paper. Furthermore, any takeaways that might exist are buried within the text, making it exceedingly difficult for the reader to identify the paper's core contributions or main message.

**Broader Impact Concerns:**

-

**Claims And Evidence:**

No

**Claims Explanation:**

Crucially, the paper does not report any performance metrics for the discussed TL methods. The paper does not provide quantitative results on (1) performance improvements of TL methods, (2) the computational efficiency of these approaches, or (3) guidelines detailing why or when a researcher should select one method over another.

Consequently, the authors fail to synthesize the current literature into any new insights, overarching conclusions, or actionable recommendations. The review currently lacks the critical analysis required to move beyond a mere summary of existing work.

**Requested Changes:**

- While the authors acknowledge the absence of existing literature reviews on transfer learning for Vision-Language Models (VLMs), they do not provide any justification for the need for one. The paper should clearly explain what unique aspects of VLM tasks or data settings necessitate specialized transfer learning approaches compared to traditional models. Why are traditional methods not sufficient?

- The paper would greatly benefit from additional statistics demonstrating the added value of the discussed methods. For instance, providing comparisons of computational efficiency or reductions in data and compute requirements for specific transfer learning techniques would strengthen the analysis. There is not a single discussion on which methods work best/when to choose them.

- All figures must be self-contained. Please improve the clarity of the visualizations and expand the captions to provide detailed explanations, ensuring the reader can understand them without referencing the main text.

-  The current tables lack any detail to be useful. They should either be significantly enhanced to convey more substantial, standalone information or removed entirely to save space.

- Currently, the paper reads primarily as an extensive summary rather than a critical review. To elevate its contribution, please include a "lessons learned" or key takeaways. Providing actionable next steps, such as recommending specific models or strategies for specific tasks, situations, or use cases, would add value for the reader. Also, there are no open question/research topics proposed unclear what are still the challanges.

---

### Review · Reviewer_nfWh · 2026-03-31

**Summary Of Contributions:**

This paper is a survey of transfer learning approaches in application to pre-trained large vision-language models (VLMs). It starts with a general overview of existing VLMs and transfer learning setups, followed by a description of most popular architectures and training objective functions in vision-language modelling. After that the authors define four categories of transfer learning approaches for VLMs: Adapter-based methods, Prompt-based methods, Model-based methods and Knowledge-base methods, and each category is accompanied with brief summaries of relevant works. The literature overview is followed by descriptions of applications and practical use cases of transfer learning for VLMs. Finally, the survey is concluded with the definitions of major vision-language tasks and the table of commonly used datasets and benchmarks categorised by the task.

**Strengths**
- The survey defines the transfer learning task for VLMs from multiple angles: network architecture, task type, objective function, method design and model evaluation which facilitates comprehensive understanding of the topic.
- The overview of transfer learning approaches in Section 4 covers a large and diverse list of relevant works, and Tables 1-5 can serve as a starting point for further studying of the literature.
- The proposed categorisation of approaches is sensible and useful for understanding the major research directions.
- The overview of applications in Section 5 highlights the importance and utility of transfer learning for VLMs for real world problems which further strengthens motivation of the paper.

**Weaknesses**
- The survey is highly "modular", i.e., it consists of sections which are completely independent and poorly connected. For example, the training objectives in Section 3 are discussed in isolation from the task definitions in Section 6, and none of these are explicitly used to describe the approaches in Section 4.
- The approaches in Section 4, in turn, are also poorly connected with each other beyond categorisation and subcategorisation. There is no comprehensive comparison or evolution of ideas behind the listed papers, and each method/paper can be removed from the survey without affecting much the surrounding text.
- Most definitions and descriptions in Sections 3, 4, 6 are high-level and minimal, and do not provide insights into individual variations and specific realisations. For example, there is no discussion about the prompt templates and the differences in input/output formats for different models and tasks. This is critical if the reader is new to the topic of vision-language modelling.
- There are no additional discussions, comparisons or analyses of the considered approaches. How do they differ in terms of performance, adaptation efficiency, number of trainable parameters, convergence speed or any other aspect? Are there any actionable insights and important trends or observations? Should certain methods be preferred over the others and what are the issues and limitations associated with each category? Unfortunately, none of these questions are addressed in the survey which significantly limits the utility of the literature overview, especially to those who are already familiar with the task.

**Audience:**

Yes

**Audience Explanation:**

For many years VLMs and transfer learning have been attracting a lot of attention. Pre-trained large VLMs can solve a number of challenging tasks involving visual and textual data, however, their size makes it difficult to adapt them to the task at hand, especially when computational resources and/or fine-tuning datasets are limited. Transfer learning applied to visual-language modelling is a straightforward solution to this problem, and an overview of such methods could potentially be of interest to a wide range of audience, including both researchers and practitioners.

At the same time it is not clear to me who is the target audience of this survey in its current state. It does not provide enough details and explanations to those who are novel to the topic, since the summaries consist of one-two sentences per paper, with many important details, definitions and illustrations omitted. At the same time there is no comparative analysis of approaches in terms of performance and/or efficiency, and there is no discussion about the properties, advantages/disadvantages, best practices and challenges that would be useful to those who already have some experience with the task.

While some readers may benefit from having an extensive list of relevant literature, categorised and provided in a single table, I have an impression that this survey alone is not sufficient to learn how to navigate among different transfer learning approaches for VLMs.

**Claims And Evidence:**

Yes

**Claims Explanation:**

Since the major part of this survey consists of paper summaries, few claims have been made. In Introduction the authors state that they "carefully summarize and classify the transfer learning methods for VLMs", as well as "study the corresponding VLMs, datasets, and realistic applications for the VLM transfer learning methods’", which they indeed do.

There is one important critique though: the title of the survey mentions "efficient adaptation", however, there is no analysis or comparison of considered approaches with respect to this characteristic. It seems as if all adaptation methods are implicitly considered to be efficient as long as they do not require training a new VLM from scratch. To me, this is a missed opportunity to discuss efficiency, an important aspect of transfer learning, which makes the title seem a bit exaggerated.

**Requested Changes:**

**Critical requests:**
- The survey would benefit from better positioning with respect to its target audience. If the goal is introduce the topic to new readers, all descriptions and definitions should be more detailed, with additional examples and illustrations for specific models. If the goal is to summarise the advances for more experiences readers, the descriptions of approaches should be accompanied with additional model comparison, analysis of insights and discussion of trends and challenges. Please refer to Major Weakness for more specific list of issues to be addressed.
- Efficiency of VLM adaptation should be explicitly discussed and quantitatively compared. Otherwise the authors may consider to remove this term from the title, since its presence might create a false impression about the content of the survey.


**Minor editing comments, suggestions and typos (do not affect recommendation):**
- Citations are managed inconsistently across the survey: in some places the lists of publications are put into brackets (e.g., on page 1), in others the lists are provided without them (e.g., the first paragraph on page 2, sentence "Several extensive papers summarize…"). This inconsistency is ubiquitous. Ideally, the lists of works should be put into brackets when the corresponding authors are not the main subjects in the sentence (as in the first two citations in the first paragraph on page 2).
- The first paragraph in Section 4 refers to "Model-based methods" as to "Pipeline Based Method", which is inconsistent with categorisation in Introduction and even with the title of Section 4.3. This may confuse readers.
- It is not clear why definition and categorisation of major vision-language tasks is discussed at the very end, in Section 6 dedicated to datasets and benchmarks. Many approaches in preceding sections are task-specific, so defining these tasks in advance would facilitate understanding of the methods.
- The phrase "…the multimodal decoders often smaller or specially-designed multimodal transformers…" is grammatically incorrect and needs reformulation (last paragraph in Section 3.1.2, page 5).
- "aims" -> "aim" (first paragraph in Section 3.1.3, page 5).
- In Table 4, some entries start with lower letters ("domain adaptation", "language understanding", "protext"), while the vast majority of other entries, as well as all entries in similar Tables 2 and 3 start with capital letters. It would be more aesthetically pleasant to keep all entries in the same formatting style.
- The entry "Cross-Modal Adaptation (Lin et al., 2023b)" is present twice in Table 4.
- "allosw" -> "allows" (first paragraph on page 14).
- "…it improves negation awareness several tasks." - a preposition might be missing in this phrase (last paragraph in Section 4.4 on page 16).
- "propose propose" -> "propose" (last paragraph in Section 5.1 on page 17).

---

### Decision · Action_Editor_o8Cw · 2026-05-13

**Recommendation:** Reject

**Additional Comments:**

N/A

**Audience:**

Yes

**Audience Explanation:**

The article can be of interest to practitioners in transfer learning. Reviewers disagreed on this point due to the lack of new material going beyond the overview of methods.

**Claims And Evidence:**

No

**Claims Explanation:**

While reviewers appreciated the overview of the approaches given by the manuscript (nfWh) on a timely topic (9DPc), they also raised several concerns on the lack of explicit criteria for inclusion, and qualitative/quantitative comparisons of the works. The authors discussed with 9DPc but did not engage with the other reviewers. All reviewers unanimously found the claims and evidence to not be supported, recommending rejection.

**Resubmission Of Major Revision:**

The authors may consider submitting a major revision at a later time.